# Comparative Life Cycle Assessment of Five Greek Yogurt Production Systems: A Perspective beyond the Plant Boundaries

**Catherine Houssard [1],\*, Dominique Maxime [1], Scott Benoit [2], Yves Pouliot [2] and Manuele Margni [1]**

[1]    CIRAIG, Mathematical and Industrial Engineering Department, Polytechnique Montréal, Technological University, Montréal, QC H3C 3A7, Canada; dominique.maxime@polymtl.ca (D.M.); manuele.margni@polymtl.ca (M.M.)

[2]    Institute of Nutrition and Functional Foods (INAF), Department of Food Sciences, Université Laval Québec, Quebec City, QC G1V 0A6, Canada; scott.benoit.1@ulaval.ca (S.B.); Yves.Pouliot@fsaa.ulaval.ca (Y.P.)

\*    Correspondence: catherine.houssard@polymtl.ca

**Abstract:** Greek yogurt (GY), a high-protein-low-fat dairy product, particularly prized for its sensory and nutritional benefits, revolutionized the North American yogurt market in less than a decade, bringing with it new sustainability challenges. Standard GY production generates large volumes of acid whey, a co-product that is a potential source of environmental pollution if not recovered. This study aims to assess the environmental performance of different technologies and identify the main factors to improve GY production. A complete life cycle assessment (LCA) was performed to compare the standard technology (centrifugation) with two new technologies (fortification and ultrafiltration) to reduce acid whey volumes. Three milk protein concentrate alternatives were also assessed. Results show that technology choice is not a clear discriminant factor. However, minimizing losses and wastage (accounting for 23 to 25% of the environmental impacts for all indicators) beyond the processing plant and selecting milk ingredients (accounting for 63 to 67% of the impacts) with low environmental impacts are key factors to improve the environmental performance of GY systems. From a methodological perspective, the results also highlight a shortcoming in the current LCA guidelines (2015) issued by the International Dairy Federation to treat the multifunctionality of GY systems.

**Keywords:** LCA; greek-style yogurt processing; environmental impacts; losses and wastage; multifunctionality; allocation

## 1. Introduction

A global transformation of the food system is urgently needed to support environmental sustainability and feed a growing population. As stated by the EAT-Lancet Commission on Food, Planet, Health, this transformation calls for a shift towards healthy diets, improved food processes and food production efficiency, and at least a 50% decrease in food losses and waste (LW) to keep humanity in a safe operating space [1]. Meat and dairy products are widely recognized as major sources of environmental and/or health burdens [2], but this statement should be qualified relative to other products. For instance, some distinguish yogurt as a sustainable food for its health benefits and low environmental impact as compared to other animal source foods. Indeed, regular yogurt production emits 75 to 55% less greenhouse gas (GHG) emissions than cheese and is a good source of calcium and protein with a lot less fat [3]. Around the world, yogurts come in various types and compositions [4]. Greek yogurt (GY), also known as Greek-style yogurt, has flooded the North

American market. In the past decade, it grew from 1% market share to around 40% in the USA [5] and 20% in Canada [6]. This spectacular success is likely attributed to GY's nutritional and sensory properties. The high-protein content provides a thick and creamy texture [7], positioning GY and positioned it as a healthy food product that provides consumers with both pleasure and nutritional benefits [8]. However, from a sustainable diet perspective, the environmental impacts of GY should also be considered. GY is recognized as a source of incremental environmental impacts in the dairy sector. Indeed, its high-protein content is generally obtained by concentrating fermented milk through centrifugation to increase the yogurt's protein content from 3.27% up to 10% *w/w*. This operation requires up to three times more milk input than regular yogurt and generates at least twice the weight of GY in acid whey [9,10]. Acid whey may be harmful to ecosystems if not recovered or disposed of correctly due to its acid pH (4.21–4.48) and high organic load (Biochemical Oxygen Demand (BOD): 45.8–50.4 mg/g) [11]. The increasing volumes of acid whey generate new operational challenges for dairy manufacturers [11]. Significant efforts have been invested to recover acid whey components. The American Chemical Society registered 3 500 patents in 2017, with most focusing on extracting proteins and lactose from GY acid whey using membrane-based filtration processes [12–15] or enzyme-based approaches [16]. However, most of these processes are not yet economically viable [5]. Therefore, instead of focusing on whey recovery, recent works have targeted processing innovation to reduce or eliminate the production of acid whey. The various technology alternatives explored are extensively described in a literature review conducted by Jørgensen et al. [8] and summarized in the supplementary materials (SM) of this paper (Section S1). These technologies influence the volume and composition of co-product generated, as well as GY composition and its sensory properties [7,17,18]. They also vary in terms of production yields, resources, utilities consumption such as energy, water use, chemicals at the manufacturing plant and capital costs of the processing equipment [8,18,19]. The technical benefits and drawbacks of each technological option are well documented but, to the best of our knowledge, no study has included a systematic environmental comparison of GY production systems and technologies.

Some life cycle assessment (LCA) studies published in the scientific literature estimate the environmental impacts of regular yogurt manufacturing process [3,20–22] and yogurt packaging and delivery systems [23,24] but none focus specifically on Greek yogurt technologies. The first aim of this paper is to fill this gap by comparing the environmental performances of the most common GY production options available in the province of Québec (Canada) in 2018 throughout the entire product life cycle from a cradle-to-grave perspective. This study also intends to identify the main contributors of GY's environmental footprint along its life cycle and key factors to reduce its environmental burden with a special focus on LW. The main findings could guide GY manufacturers' actions and priorities to improve the environmental performance of this healthy food product. In addition, from a methodological perspective, this paper aims to assess the relevance of the mass allocation method based on the dry weight of milk solid content recommended in the new LCA guidelines issued by the International Dairy Federation (IDF) [25] to partition the environmental impacts of GY from its co-products (cream and whey) and consistently compare different GY systems.

## 2. Materials and Methods

The LCA methodology is based on the common carbon footprint approach for the dairy sector [25] guideline issued by the IDF and the international ISO 14040 [26] and 14044 [27] standards. The attributional LCA compares five production alternatives that are representative of the production options available in the province of Québec (Canada) in 2018, as described in the following paragraphs. The alternatives are compared based on a functional unit of 1 kg of GY at 10% protein and 0% fat consumed by an average Québec household in 2018. Despite slight organoleptic and composition differences related to the manufacturing technology, the functional properties of the GYs compared in this study were assumed to be equivalent for the five systems.

*2.1. Compared Technological Options*

Based on a survey conducted among three major Canadian GY manufacturers in 2017 and 2018, we selected three different GY manufacturing technologies:

1.  Centrifugation (CE) is the most conventional technology. It concentrates yogurt proteins after the fermentation process stage.
2.  Fortification (FO) consists of adding protein ingredients, such as milk protein concentrate (MPC) to the milk before fermentation to reduce the quantity of acid whey generated.
3.  Ultrafiltration (UF) before fermentation is a new option that is gaining ground. It has the benefit of generating neutral pH milk permeate (instead of acid whey), which has good valorization potential on the food ingredients market.

Three ingredient supply alternatives were also considered for FO, resulting in five GY production system alternatives. The FO reference is based on solid MPC 80 powder concentrated at 80% protein (*w/w*) sourced from the USA (see description of the MPC manufacturing process in SM Section S2.2). Two alternatives to liquid diafiltered milk from the USA or Québec were also assessed.

The three MPC supply alternatives are:

1.  FO-P-US for MPC powder (P) (80% protein) from USA (US);
2.  FO-L-US for liquid (L) MPC (20% protein) from USA (US);
3.  FO-L-Qc for liquid (L) MPC (20% protein) from Québec (Qc).

*2.2. Product System Descriptions and Boundaries*

The scope of the study is cradle-to-grave. It considers all operations, from feed production to milk production at the dairy farm, yogurt factory operations, distribution, consumption and final disposal of the product. The transportation and waste management operations in all stages are also considered. All known exchanges from/to the ecosphere are included in the life cycle inventory. System boundaries are described in Figure 1.

The milk is produced by a typical Québec farm (58 cows producing on average 8137 kg FPCM/cow/year; (FPMC: fat-protein-corrected milk)) and transported to the manufacturing plant where it is transformed into GY (Figure 1).

CE: the raw milk is skimmed, thermally treated, fermented, centrifugated, and cooled before reaching the packaging area. The centrifugation process separates the acid whey from the yogurt to concentrate the proteins up to 10%.

FO: liquid or solid milk protein concentrates (MPC) are introduced between the skimming and thermal treatment operations. Other operations are similar to CE.

UF: the protein concentration is performed after skimming. Consequently, smaller volumes of skimmed milk are treated during subsequent operations. The ultrafiltration (UF) membrane retains the majority of the proteins in the skimmed milk (retentate). Most of the lactose and minerals migrate through the membrane into the permeate.

The detailed unit processes of the three technological options are described in SM Section S2.

Then, regardless of the technological option, 50% of GY is filled into 500 mL polypropylene (PP) containers and the other 50% into 100 mL polystyrene (PS) cups packed by 4 or 8 units. Packaged GY is then boxed, palletized and cooled at 4 °C to be stored. The distribution and consumption stages include several transportation and refrigeration operations, as described by [21]. In total, 79% of the cardboard used for PS packs and 15% of PP containers are recycled by consumers [28]. The rest of the materials are landfilled.

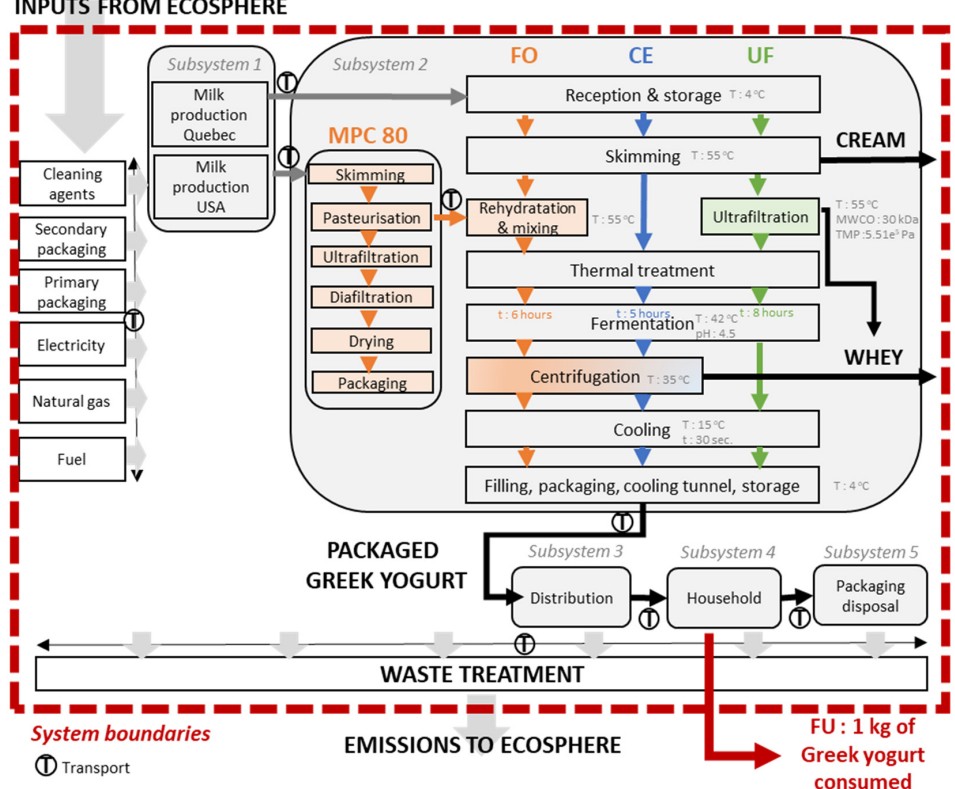

**Figure 1.** System boundaries including detailed unit process and main operating parameters for the three technologies: centrifugation (CE) in blue, fortification (FO) in orange, and ultrafiltration (UF) in green.

### 2.3. Systems Modeling Approach

We modeled the three technological options in the prototype of a dairy process simulation software [29] based on the layouts and processing parameters (temperature, pH, pressure) provided by GY manufacturers. The simulation calculated the mass balance and input and output flows per process unit. Detailed information on processing simulation parameters and input/output flow results are available in SM (Tables S1–S4). Results from the simulation were subsequently validated with GY manufacturers to improve the level of confidence of the study.

2.3.1. Mass Balance per Unit Process and Technology

The mass balances of the main process units reported in Table 1 are based on 1 kg GY output with identical protein concentration (10%). In practice, GY outputs and concentration vary according to the technology. The high selectivity of the UF membrane retains more protein and less lactose than the traditional CE. FO usually has a higher protein and lower lactose content than CE because of the high protein/lactose ratio contained in MPC 80 [8,30].

Each technology is characterized by a specific protein retention coefficient (Rp) (Equation (1)).

$$\text{Rp} = 1 - \left[P_w\right]\ \left[P_{gy}\right] \tag{1}$$

where $[P_{gy}]$ = protein concentration in GY and $[P_w]$ = protein concentration in the whey.

The simulation was performed with Rp = 0.97 ± 0.01 for CE, Rp = 0.95 ± 0.01 for FO and Rp = 0.98 ± 0.01 for UF based on manufacturer data. Output and inputs were scaled to 1 kg of GY produced, resulting in variations of raw milk input and cream and whey outputs based on the technology (Table 1).

**Table 1.** Simulated mass balance of inputs and outputs for the centrifugation (CE), fortification (FO), and ultrafiltration (UF) technologies before losses and wastage. MPC = milk protein concentrate.

| Flows | Type | CONCENTRATION ON WET BASIS | | | | | | QUANTITY PER KG OF GY PRODUCED | | | | | |
|---|---|---|---|---|---|---|---|---|---|---|---|---|---|
| | | Protein | Fat | Lactose | Ash | Dry Matter | Water | Protein | Fat | Lactose | Ash | Water | Total |
| | | (w/w) | (w/w) | (w/w) | (w/w) | (w/w) | (w/w) | kg | kg | kg | kg | kg | kg |
| Centrifugation (CE) | | | | | | | | | | | | | |
| Input | Raw milk | 3.27 | 3.97 | 4.81 | 0.75 | 12.80 | 87.20 | 0.113 | 0.138 | 0.167 | 0.026 | 3.026 | 3.47 |
| Intermediate | Skimmed milk | 3.40 | 0.04 | 5.01 | 0.78 | 9.23 | 90.77 | 0.106 | 0.001 | 0.157 | 0.024 | 2.840 | 3.13 |
| | Cream | 2.04 | 40.00 | 3.01 | 0.47 | 45.52 | 54.48 | 0.007 | 0.137 | 0.010 | 0.002 | 0.186 | 0.34 |
| Output | GY(*) | 10.00 | 0.04 | 4.66 | 0.73 | 15.43 | 84.57 | 0.100 | 0.000 | 0.047 | 0.007 | 0.846 | 1.00 |
| | Acid whey | 0.30 | 0.04 | 5.17 | 0.81 | 6.32 | 93.68 | 0.006 | 0.001 | 0.110 | 0.017 | 1.994 | 2.13 |
| Fortification (FO) | | | | | | | | | | | | | |
| Input | Raw milk | 3.27 | 3.97 | 4.81 | 0.75 | 12.80 | 87.20 | 0.089 | 0.108 | 0.131 | 0.020 | 2.381 | 2.73 |
| Intermediate | Skimmed milk | 3.40 | 0.04 | 5.01 | 0.78 | 9.23 | 90.77 | 0.084 | 0.001 | 0.123 | 0.019 | 2.235 | 2.46 |
| | Cream | 2.04 | 40.00 | 3.01 | 0.47 | 45.52 | 54.48 | 0.005 | 0.107 | 0.008 | 0.001 | 0.146 | 0.27 |
| | MPC 80 | 81.30 | 1.60 | 4.60 | 6.80 | 94.30 | 5.70 | 0.024 | 0.000 | 0.001 | 0.002 | 0.002 | 0.03 |
| | Liquid or rehydrated MPC | 24.39 | 0.48 | 1.38 | 2.04 | 28.29 | 71.71 | 0.023 | 0.000 | 0.001 | 0.002 | 0.069 | 0.096 |
| | Fortified milk | 4.20 | 0.06 | 4.87 | 0.83 | 9.96 | 90.04 | 0.107 | 0.002 | 0.125 | 0.021 | 2.304 | 2.56 |
| Output | GY | 10.00 | 0.06 | 4.56 | 0.78 | 15.40 | 84.60 | 0.100 | 0.001 | 0.046 | 0.008 | 0.846 | 1.00 |
| | Acid whey | 0.48 | 0.06 | 5.07 | 0.86 | 6.47 | 93.53 | 0.007 | 0.001 | 0.079 | 0.013 | 1.458 | 1.56 |
| Ultrafiltration (UF) | | | | | | | | | | | | | |
| Input | Raw milk | 3.27 | 3.97 | 4.81 | 0.75 | 12.80 | 87.20 | 0.111 | 0.134 | 0.163 | 0.025 | 2.949 | 3.38 |
| Intermediate | Skimmed milk | 3.40 | 0.04 | 5.01 | 0.78 | 9.23 | 90.77 | 0.104 | 0.001 | 0.153 | 0.024 | 2.768 | 3.05 |
| | Cream | 2.04 | 40.00 | 3.01 | 0.47 | 45.52 | 54.48 | 0.007 | 0.133 | 0.010 | 0.002 | 0.181 | 0.33 |
| Output | GY | 10.00 | 0.12 | 4.65 | 0.73 | 15.50 | 84.50 | 0.100 | 0.001 | 0.047 | 0.007 | 0.845 | 1.00 |
| | Sweet whey (permeate) | 0.18 | 0.00 | 5.18 | 0.81 | 6.17 | 93.83 | 0.004 | 0.000 | 0.106 | 0.017 | 1.919 | 2.04 |

(*) GY: Greek yogurt.

2.3.2. Multifunctionality Modeling Approach

GY systems are multifunctional because they produce jointly GY, cream and whey and thus fulfill several functions. Yogurt is the primary function. The cream is a high-value co-product. The acid whey from the CE or FO process is a low-value co-product recovered as an animal feed complement in pig farming, but the sweet whey (or permeate) from the UF process could be recovered for higher value applications in the food ingredient market. In the Québec context, the transport cost of acid whey to pig farms is covered by the GY manufacturer, and the pig farmer gets the whey for free. Therefore, the acid whey may be considered as a waste with a negative economic value, whereas the sweet whey from UF has a positive economic value. This difference in whey co-product functionality poses a challenge on how to allocate the impacts between the product and co-products. We used IDF guidelines [25] as a reference to allocate the impacts between the three products generated by the systems. These guidelines recommend dividing the system into subsystems when possible and using a mass allocation based on the dry weight of milk solid content in the final product and co-products when subdivision is not possible. The burdens from raw milk and manufacturing processes were then allocated between cream, GY and whey according to Equation (2):

$$\mathrm{MAF}_i \ = \ \mathrm{DM}_i \, \mathrm{Q}_i \, / \sum\nolimits_{i=1}^{n} (\mathrm{DM}_i \, \mathrm{Q}_i) \tag{2}$$

where $\mathrm{MAF}_i$ is the mass allocation on dry matter factor for product *i*; $\mathrm{DM}i$ is the dry matter content of product *i* (Table 1 column Dry Matter) and $\mathrm{Q}_i$ is the quantity of product *i* for 1 kg GY (Table 1 column total).

We also tested the robustness of the results by allocating the impacts based on an economic relationship between the co-products. This allocation was calculated based on the milk producers' revenues for class 2a milk components price average in Québec for GY and cream and class 7 milk components price average for whey in 2017. Whey received an economic allocation factor of 0% or 17.5% depending on its potential value on the market, as per LCA economic allocation rules [31].

- Scenario 1: 0% economic allocation is built on the substitution of other feed intakes on a pig farm. In this case, the whey is a cost for the GY manufacturer who pays for its transport to the pig farm.
- Scenario 2: 17.5% is built on the valorization of the whey components in the food industry for the UF process only, based on class 7 prices.

The mass and economic (Scenario 1) allocation factors are described in Table 2.

MPC manufacturing is also a multifunctional process that uses raw milk to produce cream, MPC and a UF permeate rich in valuable lactose. Economic allocation factors vary depending on the sourcing since the prices of cream and milk ingredients are not the same in the USA and Canada. We used prices according to milk components USA class IV (proteins: 3.98 USD.kg$^{-1}$; fat 5.35 USD.kg$^{-1}$; lactose 0.12 USD.kg$^{-1}$) [32], and Québec class 7 (proteins: 1.58 CAD.kg$^{-1}$; Fat 7.24 CAD.kg$^{-1}$; lactose 1.58 CAD.kg$^{-1}$) [33] in 2017. These allocation factors are available in SM (Table S6).

2.3.3. Life Cycle Inventory (LCI)

Background life cycle inventory data, including milk production, is based on the Québec inventory database [34] version 3.4 (released in 2017) in the ecoinvent database, cut-off system model (ecoinvent center https://www.ecoinvent.org/[35]) and targeted USA datasets extracted from Thoma et al. [36]. A few balance adjustments to pesticides and water flows were made to ensure the consistency of crop datasets between the USA and Québec databases. Infrastructure and processing equipment were excluded from the analyses.

**Table 2.** GY production system—mass and economic allocation factors at each point of substitution; CE (centrifugation), FO (fortification); UF (ultrafiltration); GY (Greek yogurt).

| SB = Subsystem | Mass Allocation on Dry Matter | | | | Economic Allocation (*) | | | |
|---|---|---|---|---|---|---|---|---|
| | Cream | Skimmed Milk | Whey | GY | Cream | Skimmed Milk | Whey | GY |
| **(a)** **From farm to plant (SB1)** raw milk production, transportation, losses and wastage | | | | | | | | |
| **CE** | 35% | _ | 30% | 35% | 57% | _ | 0% | 43% |
| **FO** | 35% | _ | 29% | 36% | 57% | _ | 0% | 43% |
| **UF** | 35% | _ | 29% | 36% | 57% | _ | 0% | 43% |
| **(b)** **From reception to skimming (SB2)** reception and storage, skimming | | | | | | | | |
| **CE** | 35% | 65% | _ | _ | 57% | 43% | _ | _ |
| **FO** | 35% | 65% | _ | _ | 57% | 43% | _ | _ |
| **UF** | 35% | 65% | _ | _ | 57% | 43% | _ | _ |
| **(c)** **Ultrafiltration (SB2)** | | | | | | | | |
| **UF** | _ | _ | 45% | 55% | _ | _ | 0% | 100% |
| **(d)** **From skimming to centrifugation (SB2)** MPC supply, rehydration and mixing (FO only); thermal treatment, (homogenization–optional), fermentation, centrifugation | | | | | | | | |
| **CE** | _ | _ | 47% | 53% | _ | _ | 0% | 100% |
| **FO** | _ | _ | 45% | 55% | _ | _ | 0% | 100% |
| **UF** | _ | _ | 0% | 100% | _ | _ | 0% | 100% |
| **(e)** **General plant operations (SB2)** CIP, lighting and conditioned air electricity | | | | | | | | |
| **CE** | 35% | _ | 30% | 35% | 57% | _ | 0% | 43% |
| **FO** | 35% | _ | 29% | 36% | 57% | _ | 0% | 43% |
| **UF** | 35% | _ | 29% | 36% | 57% | _ | 0% | 43% |
| **(f)** **From cooling to final disposal (SB2/SB3/SB4/SB5)** SB2 cooling, filling, packing, storage, packaging, plant wastes; SB3: transport and distribution; SB4: transport and consumption; SB5: final disposal; GY losses and wastage | | | | | | | | |
| **CE** | _ | _ | _ | 100% | _ | _ | _ | 100% |
| **FO** | _ | _ | _ | 100% | _ | _ | _ | 100% |
| **UF** | _ | _ | _ | 100% | _ | _ | _ | 100% |

(*) economic allocation is presented only for scenario 1. In scenario 2, the whey receives 17.5% of the allocation for c and d and 7.5% of the allocation for a and e. GY receives 82.5% of the allocation for c and d and 35.5% of the allocation for a and e. The economic allocation for b and f subsystems remained unchanged.

Foreground data from the process simulation were scaled up to the functional unit and summarized in Tables A1–A3 in Appendix A. Process parameters were discussed and validated with industrial partners. We collected the remaining data from the literature and tested the few assumptions with sensitivity analyses. Based on previous studies [21,37,38], we estimated electricity consumption related to packaging, refrigeration and plant ventilation and lighting to be respectively 12%, 18% and 19% for a total of 50% of the plant's total electricity consumption. We also assumed these consumptions to be identical for all options. Water flows resulting from the simulation were corrected with González-García [21] inventory to include plant general water and improve result completeness. Data from distribution and household consumption were based on González-García [21] and adapted to the Canadian context as described in SM Table S5. Refrigerant leakages were excluded, except for chilled trucks. Thoma [39] assessed the trucks at 1% of the GHG impacts at the plant and 19% during distribution, but these figures tend to drop as technology evolves. We modeled raw materials packaging recycling based on the cut-off approach using current recycling rates in Québec as recycled content rates. Milk and GY L and W through the value chain, including packaging and all related operations (processing, storage, refrigeration and transportation), except disposal, were assessed at 3.5% during milk production and transportation, 3% at the processing plant and 26% between distribution and consumption based on a literature overview described in SM (Table S7). SimaPro v8.5.2.2 software [40]

was used to model the LCI. SM (Table S5) provides key parameters and the sources used to build each operation.

### 2.4. Life Cycle Impact Assessment

Impacts are estimated from the LCI using the IMPACT WORLD + method [41], Endpoint v1.4.1, and Midpoint v1.23. Four impact category indicators were selected to compare the GY system options providing a high level of aggregation of environmental concerns relevant to Québec's dairy sector.

- Human health (HH) in DALY(*) encompassing the impact on water availability, human toxicity, particulate matter formation, etc.
- Ecosystem quality (EQ) in pdf·m$^2$·yr encompassing the impacts on freshwater eutrophication, land occupation, land transformation, etc.
- Climate change (CC) in kg $CO_2$ eq.
- Fossil and nuclear energy use (FEU) in MJ deprived

(*) Disability-adjusted life year

Climate change is measured in the short term, representing a time integrated impact over a time horizon of 100 years. The contribution of climate change to the HH and EQ endpoint indicators was removed purposely. For the raw milk data from the USA imported from an older version of ecoinvent (v2.2), water supply processes were adapted to ensure the water mass balance. However, the water balance is not consistently ensured for the background processes across ecoinvent v2.2. Therefore, the representativity of the water availability impact assessment in the HH and EQ categories should be considered carefully.

## 3. Results

### 3.1. Contribution to the Environmental Impacts

Results presented in Figure 2 are expressed in absolute value. Detailed numerical results are available in SM (Sections S7.1 and S7.2). The cradle-to-grave perspective reveals the significant contributions of milk production, milk ingredients (MPC) and product LW to the GY impacts. Packaging materials are also important contributors when it comes to non-renewable energy use.

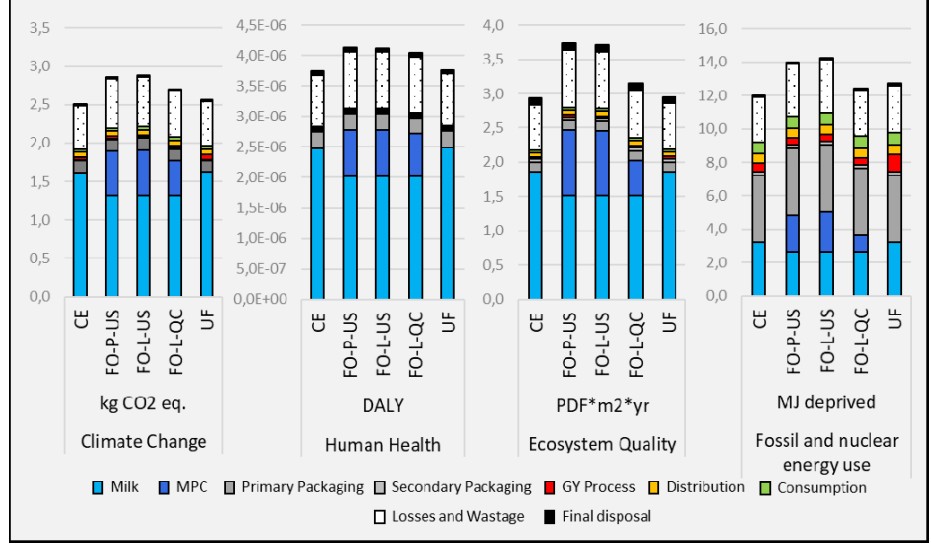

**Figure 2.** Comparative LCA of centrifugation (CE), ultrafiltration (UF), and fortification (FO) with 3 MPC sourcing alternatives (FO-P-US: fortification with MPC 80 powder from USA; FO-L-US: fortification with liquid MPC from USA; FO-L-Qc: fortification with liquid MPC from Québec) for a functional unit (FU) of 1 kg yogurt consumed including LW.

The following section details the results in Figure 2 in %.

Milk production at the farm (including milk ingredients for FO) contributes to 82 to 88% of the climate change (CC), human health (HH), and ecosystem quality (EQ) impacts and to 33 to 43% of the fossil energy use (FEU) impacts for the five production alternatives before LW. This is consistent with the IDF report (2009) [42], which states that milk production contributes to 80 to 95% of the total LCA GHG emissions based on 60 dairy products studies. When LW is included, the main contributors are still milk and dairy ingredients, which contribute to 63 to 67% of the CC, HH, and EQ impacts and 25 to 35% of the FEU impacts.

Losses and wastage (LW) contribute to 23 to 25% of the life cycle impacts for all impact indicators. Potential impacts from LW are notably higher than the cumulative impacts of the processing, distribution, consumption and final disposal stages.

Primary packaging, which mainly consists of PP and PS containers, contributes to 6 to 8% of CC impacts and to 35% in the FEU category (Figure 2). For half of the GY volumes packaged in PP and half in PS, the latter requires 24% more raw material (Table A1) and has respectively 25% and 50% more impacts on FEU and CC than the former. The contribution of packaging to CC (6%) remains low as compared to the contribution of milk and milk ingredients (>63%) or LW (>24%).

Processing operations at the plant contribute to less than 3% of CC, HH, EQ, and between 3 to 8% of FEU impacts over the entire life cycle for all production alternatives. Operations requiring the use of natural gas (heating treatment and Clean-in-place (CIP)) stand out in the FEU category. They contribute to 94 to 98% of the processing stage impacts. More details on the contribution of each processing operation are available in SM (Table S9).

Distribution and consumption stages include mainly electricity consumption for refrigeration and fuel consumption for transportation. The sum of the operations accounts for 4% of CC impacts and 10% of FEU impacts over the entire life cycle in this study (Figure 2) but may vary significantly depending on transportation distances. For instance, the contribution to CC impacts varies from 1 to 18% over the entire life cycle (1% for transport from the grocery to home on foot and distribution distances < 50 km; 18% being for 20 km from the grocery by car and distribution distance equal to 600 km).

Final disposal accounts for less than 1% of CC, 2% of HH, 3.5% of EQ, and 1% of FEU impacts over the entire GY life cycle.

### 3.2. Comparison of LCA Production Alternatives

The relative discrepancy between the highest and lowest environmental impact scores among production alternatives varies from 0 to 20% across indicators. The two fortification alternatives sourced in the USA have the greatest impacts for all four indicators. CE has the lowest impact. The discrepancies for CE, UF, and FO-L-QC are less than 8% for each indicator.

### 3.2.1. Protein Yield and Raw Milk Input

Seeing as the major contributor to the environmental impacts is milk production (Figure 2), the technologies that consume less milk to achieve 10% protein concentration in GY were expected to perform better. So, the higher the protein yield performance, the more GY is produced with the same quantity of milk and the more the impacts correlated to GY production may decrease. As reported in Table 1, FO is the most intensive technology in terms of milk input (3.51 per kg of GY produced including 2.73 kg of QC raw milk and 0.78 kg of milk for MPC), followed by CE (3.47 kg) and UF (3.38 kg). FO has a low protein retention rate in GY during the centrifugation process (Rp = 0.95). The performance of UF is attributed to the good selectivity of the membrane, which retains more protein in the GY (Rp = 0.98) than the CE separator (Rp = 0.97). Nevertheless, LCA results are not only correlated with protein yield. Several other factors are involved, such as the influence of the milk producing region and the type of MPC (liquid or powder) used. These key factors are discussed below, and other, less influential, factors are detailed in SM Section S7.3.

### 3.2.2. Influence of the MPC Milk Producing Region and Type

Influence of the milk producing region (Figure 3): the discrepancies between the three MPC sourcing alternatives (FO-S-US, FO-L-US and FO-L-QC) for the FO process (Figure 3) are due to a combination of three main factors: milk producing region, MPC drying process, and transportation distances.

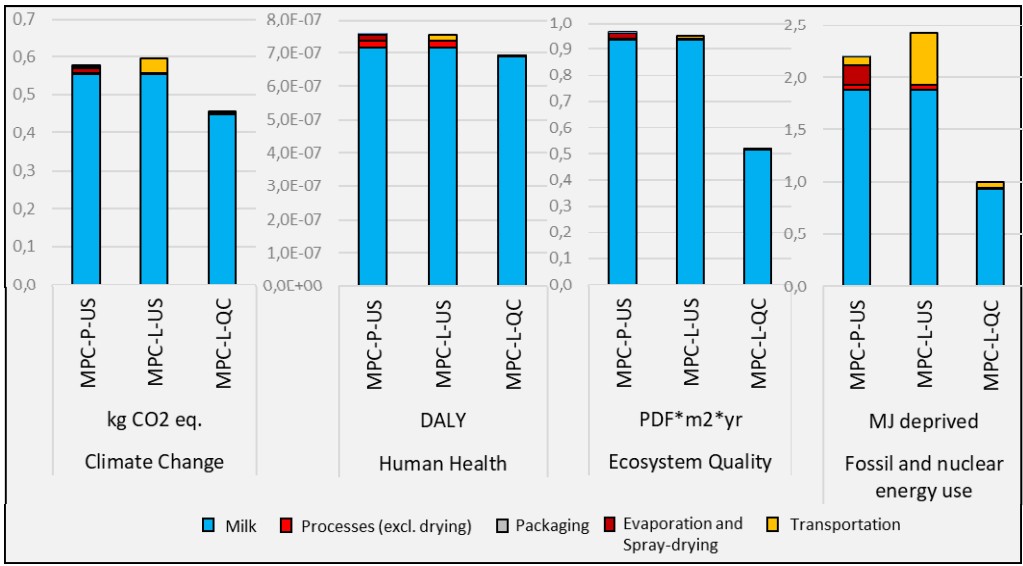

**Figure 3.** Impact profile for the production and transportation of three MPC sourcing alternatives with quantities scaled up to fulfill the functional unit (1 kg of GY). MPC-P-US: 0.029 kg of MPC 80 powder from the USA transported over 1500 km; MPC-L-US: 0.12 kg of liquid MPC from the USA transported over 1500 km; MPC-L-Qc: 0.12 kg of liquid MPC from Québec transported over 180 km.

Milk production contributes to 77 to 99% of the MPC impacts for all the FO alternatives and impact categories. Milk produced in the USA (national average) has a greater impact than milk produced in Québec by 19% for CC, 4% for HH, 45% for EQ, and 56% for FEU. Differences in farming system parameters such as feed intakes, crop production practices, irrigation requirements, manure management and regional climate conditions, lead to most of these variations [43]. On average, the USA's agricultural practices are more intensive than Canada's. The USA uses more fertilizers, requires more fossil energy and relies on more maize for cow feed (a crop with a high environmental footprint) than Canada [44]. Furthermore, a sensitivity analysis based on data collected by Thoma et al. [36] on American farms reveals significant variations between USA regions (SM Table S10).

Influence of the MPC drying process and transportation distances (Figure 4): for CC and FEU impact categories, the supply of 1 kg of MPC powder (MPC-P-US) transported over 1500 km has respectively only 4 and 9% fewer impacts than the supply of 4 kg of liquid MPC (MPC-L-US). By reducing transportation distances to 750 km or less, the supply of 1 kg of MPC powder (MPC-P-US) has more impacts than the supply of 4 kg of liquid MPC across all categories. As already demonstrated by Depping et al. [45], transporting large volumes of liquid proteins over long distances often has fewer impacts than drying operations. As shown in Figure 4 and detailed in SM Section S8.3, the milk sourcing region and type of MPC (powder versus liquid) are much more sensitive parameters than transportation distances.

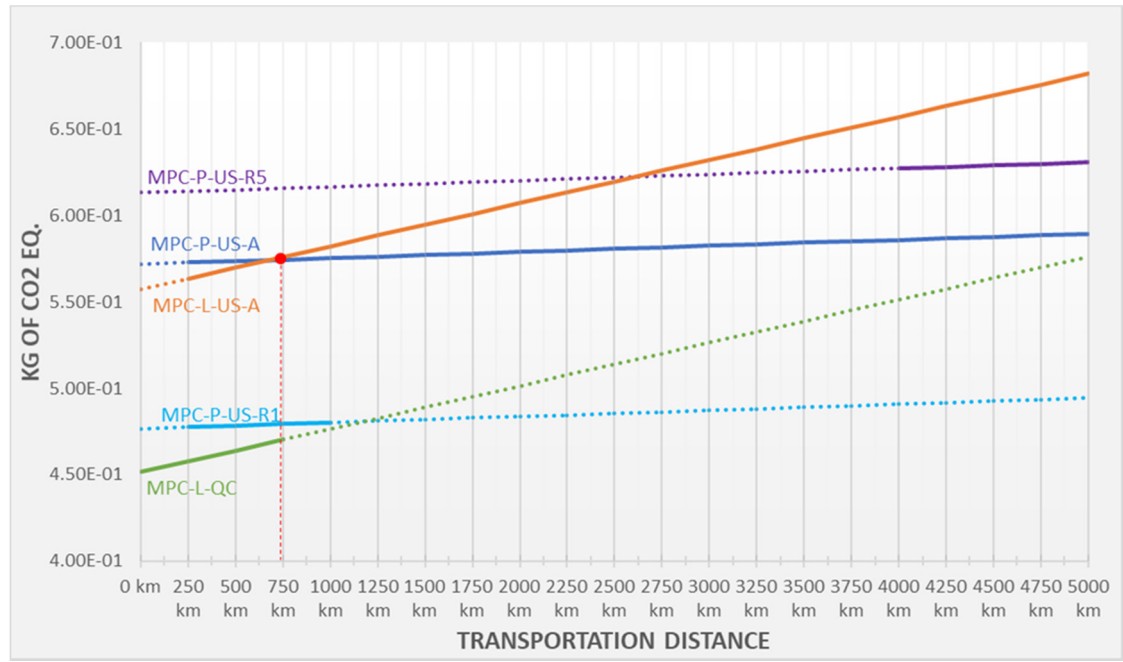

**Figure 4.** Variation in CC impact as a function of the transportation distance from MPC plant to GY plant for the three MPC sourcing alternatives scaled to the FU (1 kg of GY): MPC-P-US-A: 0.03 kg MPC 80 powder from USA with USA raw milk average; 0.12 kg MPC-L-US -A: liquid MPC from USA with USA raw milk average; 0.03 kg MPC-P-US -R1: MPC 80 powder from north east USA; 0.03 kg MPC-P-US -R5 MPC 80 powder from west coast USA; 0.12 kg MPC-L-Qc: liquid MPC from Québec.

### 3.3. Results Sensitivity

The different sensitivity analyses performed on the modeling parameters or the modeling and methodological choices (detailed in SM Sections S8.1 to S8.4) show that the results are mainly influenced by the quantity and origin of supplied raw milk or MPC, allocation rules and factors and LW rates.

Economic versus mass allocation on dry matter: as summarized in Table 3, the conclusions are sensitive to the allocation rules (mass versus economic) and allocation factors (value attributed to the whey). With economic allocation (Scenario 1), the whey is considered a waste and attributed 0% of the impacts. With mass allocation on dry matter, the whey (or permeate) is attributed 29 to 30% of the raw milk impacts for UF and CE, respectively. This difference changes the impacts allocated to GY and modifies the conclusions between production alternatives. Indeed, with economic allocation, the higher protein retention coefficient of UF gives UF an advantage as compared to CE by slightly reducing the mass of raw milk required at the input (3.47 kg/kg GY for CE and 3.38 kg/kg GY for UF) to produce 1 kg of GY. However, with mass allocation on dry matter, this advantage is offset by the higher dry matter retention rate of UF versus CE (Table 1), which increases the impacts allocated to GY processed by UF versus CE (Table 2).

If the milk components from the UF permeate is recovered (scenario 2 of the economic allocation) instead of being treated as waste like acid whey (Scenario 1 of the economic allocation), UF performs even better than CE and FO. Indeed, in this situation, only 82.5% of the UF system impacts are allocated to GY, whereas GY made from CE and FO systems still received 100% of the impacts.

Milk sourcing region: the conclusions are also sensitive to the MPC milk sourcing region (Table 3 and SM Table S16). For instance, if MPCs are produced with USA milk sourced from the state of New York (R1), FO-L-US performs better than FO-P-US in all impact categories. In contrast, if MPCs are produced with USA milk sourced from California (R5), FO-P-US performs better than FO-L-US. FO-L-QC performs better than the FO-US alternatives, but CC impact results show slight gaps (0.2%) with milk production regions such as R1 and major substantial gaps (14%) with milk production regions such as R5 (Figure 4 and SM Section S8.4 Table S16). These findings reinforce the observation

that the region from which the milk and milk ingredients are sourced has a major influence on GY's environmental performance.

LW rate reduction potential: a sensitivity analysis (detailed in SM Section S8.4, Figure S4) shows that a 1% reduction in LW at the GY plant is more effective than a 10% reduction in energy consumption to reduce CC impacts. The LW from distribution to consumers (20% of the life cycle impacts) have an even greater impact mitigation potential.

**Table 3.** Changes in production alternative classification according to sensitivity analyses. Changes compared to the reference are identified in blue. CC: climate change; HH: human health; EQ: ecosystem quality; FEU: fossil energy use; CE: centrifugation, UF: ultrafiltration; FO-P-US: fortification with MPC 80 powder from USA; FO-L-US: fortification with liquid MPC from USA; FO-L-Qc: fortification with liquid MPC from Québec: R1: north east; R5: west coast.

| OBJECT | MODIFICA-TION | IMPACT CATEGORY | LCA RESULTS | CONCLUSION VERSUS REFERENCE | GENERAL CLASSIFICATION |
|---|---|---|---|---|---|
| Reference | NA | CC | CE < UF < FO-L-QC < FO-P-US < FO-L-US | | CE < UF < FO-L-QC except for FEU FO alternatives variable |
| | | HH | CE < UF < FO-L-QC < FO-L-US < FO-P-US | | |
| | | EQ | CE < UF < FO-L-QC < FO-L-US < FO-P-US | | |
| | | FEU | CE < FO-L-QC < UF < FO-P-US < FO-L-US | | |
| Allocation | Economic rather than mass allocation on dry matter | CC | FO-L-QC< UF < CE <FO-P-US<FO-L-US | Changed | **FO-L-QC< UF< CE** Except for FEU others vary |
| | | HH | FO-L-QC<UF< CE<FO-L-US=FO-P-US | Changed | |
| | | EQ | FO-L-QC< UF<CE <FO-L-US <FO-P-US | Changed | |
| | | FEU | FO-L-QC<CE<UF< <FO-L-US <FO-P-US | Changed | |
| | Economic allocation with whey UF at 17.5% rather than 0% | CC | UF< FO-L-QC < FO-L-US<FO-P-US< CE | Changed | UF<FO-L-QC Except for FEU others vary |
| | | HH | UF< FO-L-QC <CE <FO-L-US<FO-P-US | Changed | |
| | | EQ | UF <FO-L-QC<CE <FO-L-US <FO-P-US | Changed | |
| | | FEU | FO-L-QC<UF<CE <FO-L-US <FO-P-US | Changed | |
| US milk sourcing region | R1 350 km vs. national average of 1500 km | CC | CE<UF<FO-L-QC <FO-L-US<FO-P-US | Changed | CE lowest and FO-L-QC<FO-L-US<FO-P-US |
| | | HH | CE<UF<FO-L-QC <FO-L-US<FO-P-US | Unchanged | |
| | | EQ | CE<UF<FO-L-QC <FO-L-US<FO-P-US | Unchanged | |
| | | FEU | CE<FO-L-QC<UF <FO-L-US <FO-P-US | Changed | |
| | R5 5000 km vs. national average 1500 km | CC | CE<UF<FO-L-QC <FO-P-US<FO-L-US | Unchanged | CE lowest and FO-L-QC<FO-P-US <FO-L-US |
| | | HH | CE<UF<FO-L-QC<FO-P-US<FO-L-US | Changed | |
| | | EQ | CE <UF<FO-L-QC <FO-P-US <FO-L-US | Changed | |
| | | FEU | CE<FO-L-QC <UF <FO-P-US <FO-L-US | Unchanged | |

## 4. Discussion

### 4.1. Key Findings for GY Manufacturers

This study demonstrates that milk production is the most important contributor to the environmental profile of GY. Any initiatives from manufacturers to select milk from regions with a low environmental footprint and reduce milk inputs and product LW could significantly improve the environmental performance of GY. Manufacturers could also promote and support efforts to mitigate the impacts of dairy farms.

Procurement decisions about milk and milk protein sourcing significantly impact the performances of GY systems. Depending on transportation distances, the choice of liquid protein ingredients and protein powders can improve the environmental performances of GY. However, the milk production region remains the most important factor. The large variations observed among regions are mainly due to differences in farming practices, such as crop feeding and manure management, and, to a lesser extent, regional climate. Therefore, any efforts by manufacturers to improve the traceability and impact assessment of their supply chains (MPC and milk sourcing) could have a positive influence on the environmental profile of GY.

Waste minimization has already been identified as a priority in some LCA studies in dairy literature [20,46–49]. Our results reinforce the idea that reducing LW along the life cycle is an effective lever to improve the environmental impact of GY and should be investigated further. We demonstrated that reducing LW by 1% at the plant level is more effective than reducing energy consumption by 10% in the Québec context and could be even more effective if the initiative extends beyond the plant. For instance, cooperating with food security legislators to improve consumer communication on product shelf life labeling and participating in public awareness campaigns on the environmental impact of dairy food wastage could influence the reduction in losses in the distribution and consumption stages. Instead of being viewed as a potential factor that lowers sales volumes, this type of reduction could become a factor of competitive differentiation and commitment in terms of corporate social responsibility.

In line with the LCA literature on regular yogurt [3,21], manufacturers that want to improve the efficiency of their plant operations should focus on improving heat exchangers and CIP energy consumption, which are the major contributors to plant's environmental impacts. However, in contrast to the results for regular yogurt, the energy consumption at the manufacturing plant is not a critical factor over the entire product life cycle of GY and the differences in energy consumption between each technology are not discriminating. The contribution of the processing operations stage (3% of CC impacts for all alternatives) is very low as compared to the literature on standard yogurt, which accounts for around 40% of CC impacts [3,21]. This is explained by three factors: (1) milk input for GY, which is up to three times greater than regular yogurt; (2) the consideration of LW separately from the life cycle stages in this specific study (3) the electricity mix of Québec (mostly hydropower), which has an extremely low carbon footprint as compared to other regions of the world [50]. The impacts related to processing operations will increase in regions where fossil thermal power plants are the primary source of electricity. However, with the electricity mix average of the USA (35% natural gas; 27% coal; 19% nuclear; 17% renewable [51]) the impacts related to CC rise to only 2%, thus, demonstrating that the contribution of the processing stage to the entire GY life cycle impacts remains low in any case.

Finally, it is difficult to distinguish the environmental performance of each technology (CE, FO, UF) based on these results. Indeed, the results are sensitive to the allocation rules used between GY and its co-products and the milk sourcing regions. While CE tends to be the preferred alternative with mass allocation on dry matter, UF performs best with economic allocation.

*4.2. Limitations and Opportunities*

This study paves the way for a range of opportunities for further research. The major limitation pertains to the multifunctionality approach, which is a highly sensitive choice when determining the environmental burdens of various co-products. The accuracy of the LW data is another area of improvement.

4.2.1. Multifunctionality and Allocation Method

Comparative LCA results between the five studied systems are sensitive to the allocation rules. While the sweet whey from the UF technology has a good potential to be recovered in the food industry, the acid whey from the CE and FO technologies is still considered a by-product and managed as waste. In this respect, the IDF allocation based on dry-matter content is somewhat misleading in this

comparative LCA. This rule attributes more impacts to the product that retains more dry matter in GY. Consequently, 1% more impacts from the raw milk (Table 2) are allocated to GY produced with the UF technology despite generating a co-product that is more valuable than the acid whey from CE or FO. Furthermore, fewer impacts are allocated to GY with the technologies that reject more valuable milk components in the acid whey. Reducing GY impacts by allocating the environmental burdens to the acid whey is hardly a justifiable choice when the results are to be used in an environmental declaration initiative. Therefore, despite ISO recommendations to prioritize physical causal relationships (such as dry-matter content) in allocation methods (ISO 14044) [27], in the current situation in which GY manufacturers are still struggling to recover the milk solid content that ends up in the acid whey, we would recommend an economic allocation zeroing the allocation to acid whey. However, economic allocations depend upon price volatility and market demand. With this rule, results and conclusions may vary over time. For instance, the sensitivity analysis on the economic allocation value of whey (Table 3) showed that the fluctuations in the prices of UF whey components alter the conclusions. UF may score better or worse as compared to the other technologies depending on the market value of the permeate.

Dealing with multifunctionality has always been an issue in LCA [52]. Many different solutions to the allocation problems have been suggested but it has been demonstrated in this study and others that choosing one allocation rule can have a decisive impact on the LCA results [48,53,54]. Recent LCA dairy studies [37,55,56] using the allocation approach recommended by the IDF also found that the allocation section of the guideline should be improved. Our work demonstrated that mass allocation on dry matter is not the most appropriate choice for GY and that economic allocation is not a panacea. Further work to assess other approaches that address the multifunctionality of GY and, more generally, dairy production is therefore recommended but is beyond the scope of this paper.

### 4.2.2. LW Data Improvement

We used available data in the literature to assess the environmental burdens (SM Section S6) of LW but sources are limited and often based on assumptions. The amount of milk and GY LW along the supply chain were accounted for when calculating the reference flows. However, the treatment of these wastes along the supply chain was not considered due to the lack of information on the different waste treatment pathways and technologies involved.

Work to improve the accuracy of LW data is not trivial. Since it is a major contributor to the entire product life cycle, any reduction in this area could lead to a significant improvement in the environmental performances of dairies.

### 5. Conclusions

This first comprehensive LCA comparing three GY producing technologies (CE, FO, UF) and three different MPC sourcing alternatives concluded that it is impossible to clearly discriminate the environmental performance between the assessed production alternatives. The conclusions remain sensitive to the allocation modeling choices between GY and its co-products. Nevertheless, our findings identify the environmental hotspots across the GY value chain—something that has never been done before. It comes as no surprise that the major contributor is raw milk production, and the study shows that the GY manufacturing stage is a low contributor to the environmental burdens of GY systems. Minimizing product LW across the entire value chain (estimated at 32.5% overall) and sourcing milk from regions with the lowest environmental impacts could significantly improve GY performances. Therefore, LW minimization should remain a key priority at the manufacturing plant. In addition, GY manufacturers can also significantly influence the life cycle performance of GY systems beyond their plant operations through targeted strategic decision-making related to ingredient procurement, product shelf life and returned product management. This broader life cycle vision beyond the plant boundaries is a step forward to guide the dairy industry toward a sustainable path.

Beyond these findings, the study also contributes to the debate on defining the most appropriate modelling to solve the multifunctionality of dairy systems. We demonstrated the shortcomings of applying the mass allocation method based on the dry matter weight recommended by the IDF to compare GY systems producing yogurt, cream, and whey with different characteristics. Further research is required to gain a better understanding of whey valorization pathways and determine more holistic approaches to model such multifunctional systems.

**Supplementary Materials:** The following are available online at http://www.mdpi.com/2071-1050/12/21/9141/s1.

**Author Contributions:** Conceptualization, C.H.; methodology, C.H., S.B., and D.M.; software, S.B.; validation, M.M., D.M., and Y.P.; formal analysis, C.H.; investigation, C.H., S.B.; resources, Y.P; writing—original preparation, C.H., writing—review and editing, M.M., S.B., D.M., Y.P.; supervision, M.M. and Y.P.; project administration, M.M.; funding acquisition, Y.P. All authors have read and agreed to the published version of the manuscript.

**Funding:** This research was funded by Novalait and the Fonds de recherche du Québec—Nature et technologies (FRQNT).

**Acknowledgments:** We would like to express our gratitude to Novalait, the Fonds de recherche du Québec—Nature et technologies (FRQNT), Institut EDDEC and TD Bank for their financial support. We would also like to thank Greg Thoma and his research team at the University of Arkansas, who provided us with the USA LCI datasets, Producteurs de lait du Québec, and the three Canadian yogurt manufacturers involved in this project for their contributions. Their valuable collaboration in selecting the technical processing options, collecting foreground data, and validating the results significantly improved the quality of our work.

**Conflicts of Interest:** The authors declare no conflict of interest. The funders had no role in the design of the study; in the collection, analyses, or interpretation of data; in the writing of the manuscript, or in the decision to publish the results.

# Appendix A

**Table A1.** Global inventory scaled up to the functional unit (FU): 1 kg of GY at 10% protein and 0% fat consumed by an average Québec household in 2018 before LW. CE: centrifugation; FO-P-US: fortification with MPC 80 powder from the USA; FO-L-US: fortification with liquid MPC from the USA; FO-L-QC: fortification with liquid MPC from Canada; UF: ultrafiltration.

| | Unit | CE | FO-P-US | FO-L-US | FO-L-QC | UF | Source |
|---|---|---|---|---|---|---|---|
| **Inputs from Technosphere** | | | | | | | |
| *Raw material procurement (SB1)* | | | | | | | |
| Raw milk | kg | 3.47 | 2.73 | 2.73 | 2.73 | 3.38 | SM Simulation Benoit and Houssard (Tables S1 to S4) and Table S5 |
| MPC powder | kg | – | 0.030 | – | – | – | |
| MPC liquid | kg | – | – | 0.119 | 0.119 | – | |
| Culture of lactic ferments (not included) | | NI | NI | NI | NI | NI | |
| Raw milk transportation to Qc plant | t·km | 0.654 | 0.516 | 0.516 | 0.516 | 0.637 | PLQ (2016) |
| MPC transportation to GY plant | t·km | – | 0.044 | 0.178 | 0.018 | – | SM Table S5 |
| *Primary packaging (SB2)* | | | | | | | |
| PP containers (50% of FU)—Polyethylene (virgin content) | g | 15.025 | 15.025 | 15.025 | 15.025 | 15.025 | SM Table S5 |
| PP containers recycled content | g | 2.640 | 2.640 | 2.640 | 2.640 | 2.640 | |
| PS containers (50% of FU) Polystyrene | g | 23.041 | 23.041 | 23.041 | 23.041 | 23.041 | |
| PET seal for PP containers | g | 0.512 | 0.512 | 0.512 | 0.512 | 0.512 | |
| Laminated paper seal for PS containers | g | 1.200 | 1.200 | 1.200 | 1.200 | 1.200 | |
| HDPE lid for PP containers | g | 7.172 | 7.172 | 7.172 | 7.172 | 7.172 | |
| Bleached cardboard for PS containers (virgin content) | g | 3.230 | 3.230 | 3.230 | 3.230 | 3.230 | |
| Bleached cardboard for PS containers (recycled content) | g | 12.320 | 12.320 | 12.320 | 12.320 | 12.320 | |

**Table A2.** Global inventory scaled up to the functional unit (FU): 1 kg of GY at 10% protein and 0% fat consumed by an average Québec household in 2018 before LW (cont'd).

| | Unit | CE | FO-P-US | FO-L-US | FO-L-QC | UF | Source |
|---|---|---|---|---|---|---|---|
| **Inputs from Technosphere** | | | | | | | |
| *Secondary packaging (SB2)* | | | | | | | |
| Corrugated board | g | 48.860 | 48.860 | 48.860 | 48.860 | 48.860 | |
| LLDPE stretch wrap film | g | 0.788 | 0.788 | 0.788 | 0.788 | 0.788 | SM Table S5 |
| Wood pallet | g | 0.141 | 0.141 | 0.141 | 0.141 | 0.141 | |
| *GY Processing at plant (SB2)* | | | | | | | |
| *Electricity* | | | | | | | |
| Milk filling and storage at 4 °C | Wh | 0.136 | 0.107 | 0.107 | 0.107 | 0.133 | |
| Heating at 55 °C | Wh | 0.156 | 0.127 | 0.127 | 0.127 | 0.247 | |
| Skimming at 55 °C | Wh | 3.611 | 2.842 | 2.842 | 2.842 | 3.520 | |
| Fortification at 55 °C | Wh | _ | 0.830 | 0.077 | 0.077 | _ | SM Simulation Benoit and Houssard (Tables S1 to S4) |
| Ultrafiltration at 55 °C | Wh | _ | _ | _ | _ | 1.011 | |
| Thermal treatment at 88 °C for 6 min. | Wh | 1.440 | 1.350 | 1.350 | 1.350 | 1.041 | |
| *Homogenization at 65 °C and 170–200 bars (optional)* | *Wh* | *17.400* | _ | _ | _ | _ | |
| Cooling at 42 °C | Wh | 0.247 | 0.208 | 0.208 | 0.208 | 0.041 | |
| Fermentation at 42 °C during 5 to 8 h | Wh | 0.289 | 0.245 | 0.245 | 0.245 | 0.044 | |
| Centrifugation at 35–40 °C | Wh | 15.053 | 11.847 | 11.847 | 11.847 | _ | |
| Cooling at 15 °C in 20 to 30 sec. | Wh | 3.102 | 3.538 | 3.538 | 3.538 | 3.059 | |
| CIP | Wh | 0.167 | 0.164 | 0.164 | 0.164 | 0.162 | |
| Packaging and storage at 4 °C | Wh | 14.236 | 14.236 | 14.236 | 14.236 | 14.236 | [21,38] |
| Plant ventilation and lighting | Wh | 9.491 | 9.491 | 9.491 | 9.491 | 9.491 | [21,38] |
| *Natural gas* | | | | | | | |
| Heating treatments regeneration system | MJ | 0.618 | 0.490 | 0.490 | 0.490 | 0.738 | SM Simulation Benoit and Houssard (Tables S1 to S4) |
| CIP | MJ | 0.087 | 0.086 | 0.086 | 0.086 | 0.085 | |
| *Chemicals and water* | | | | | | | |
| Sodium hydroxide in 50% solution state | g | 0.361 | 0.356 | 0.356 | 0.356 | 0.351 | SM Simulation Benoit and Houssard (Tables S1 to S4) |
| Nitric acid in 50% solution state | g | 0.139 | 0.137 | 0.137 | 0.137 | 0.135 | |

**Table A3.** Global inventory scaled up to the functional unit (FU): 1 kg of GY at 10% protein and 0% fat consumed by an average Québec household in 2018 before LW (continued and end).

| | Unit | CE | FO-P-US | FO-L-US | FO-L-QC | UF | Source |
|---|---|---|---|---|---|---|---|
| **Inputs from Technosphere** | | | | | | | |
| *Chemicals and water* | | | | | | | SM Simulation Benoit and Houssard (Tables S1 to S4) |
| Deionized water for MPC powder hydration | kg | – | 0.090 | – | – | – | |
| Deionised water for CIP | kg | 0.255 | 0.252 | 0.252 | 0.252 | 0.249 | |
| Other plant tap water usage | kg | 2.941 | 2.315 | 2.315 | 2.315 | 2.867 | [21] |
| Distribution (SB3) | | | | | | | |
| Electricity | Wh | 186.100 | 186.100 | 186.100 | 186.100 | 186.100 | [21] calculated |
| Transportation | t·km | 0.145 | 0.145 | 0.145 | 0.145 | 0.145 | SM Table S5 |
| Consumption (SB4) | | | | | | | |
| Plastic bag | g | 2.000 | 2.000 | 2.000 | 2.000 | 2.000 | [57] calculated |
| Transportation | km | 0.146 | 0.146 | 0.146 | 0.146 | 0.146 | SM Table S5 |
| Electricity (refrigeration) | Wh | 54.700 | 54.700 | 54.700 | 54.700 | 54.700 | [21] |
| Tap water | kg | 0.8045 | 0.8045 | 0.8045 | 0.8045 | 0.8045 | [21] |
| **Output to technosphere** | | | | | | | |
| Wastes to treatment (SB2, SB3, SB4, SB5) | | | | | | | |
| White water from plant | m3 | 3.20E-03 | 2.57E-03 | 2.57E-03 | 2.57E-03 | 3.12E-03 | SM Simulation Benoit and Houssard (Tables S1 to S4) Calculation SM Table S5 |
| Other waste water treatment | m3 | 8.05E-04 | 8.05E-04 | 8.05E-04 | 8.05E-04 | 8.05E-04 | |
| Cardboard and corrugated board | g | 71.160 | 71.160 | 71.160 | 71.160 | 71.160 | |
| Plastic mixture landfill | g | 49.178 | 49.178 | 49.178 | 49.178 | 49.178 | |
| Municipal waste collection (transportation) | t·km | 1.19E-02 | 1.13E-02 | 1.13E-02 | 1.13E-02 | 1.13E-02 | |
| Product and co-products (SB2) | | | | | | | |
| Cream | kg | 0.341 | 0.268 | 0.268 | 0.268 | 0.332 | SM Simulation Benoit and Houssard (Tables S1 to S4) |
| **Greek Yogurt (GY)** | kg | **1.000** | **1.000** | **1.000** | **1.000** | **1.000** | |
| Whey | kg | 2.129 | 1.559 | 1.559 | 1.559 | 2.044 | |

S4: See extensive details in Table S5 of the supplementary materials file.

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
