# Peer review of "Comparative Life Cycle Assessment of Five Greek Yogurt Production Systems: A Perspective beyond the Plant Boundaries"

_sustainability, doi:10.3390/su12219141_

Round 1
Reviewer 1 Report
This work is a significant contribution to the LCA-mapping of dairy product manufacturing, it gives a valuable overview over the huge machinery that is working to give consumers their desired products. Substantial effort is done to map energy inputs during production and follow the waste-lines all through the life span of the product. At some point the data collected are almost too much and overwhelming with all the background data given in the supplement. And at some points it is hard to follow the story, due to all the factors and input numbers.
Attached is a detailed list of comments and suggestions to the article.

Author Response
Thank you very much for having taken the time to do an extensive revision of our manuscript. You made valuable comments to improve the quality of the manuscript. Please find below our answers to your comments.
PLEASE SEE ATTACHMENT.

Reviewer 2 Report
The aim of the research was to assess the environmental performance of different technologies and identify the main factors for improving greek yogurt production. Study have shown that the technology choice is not a clear discriminant factor. However, minimizing losses and wastage beyond the processing plant and selecting milk ingredients with low environmental impacts are key factors in improving the environmental performance of GY systems.
A large - scale research was done, and the scientific value of the study is unquestionable. The research results are interesting not only for scientists, but also for anyone interested in environmental sustainability issues.
The manuscript is original, results are scientificly significant, conclusions justified and supported by the results.
There are some observations that should be noted.
The information you provided in lines 77-79 is not required.
Line 163-164 "Table 1. Simulated mass balance of inputs and outputs for the CE, FO, and UF technologies before L&W". There is need explanation for the abbreviation L&W in the text. You wrote explanation in line 267.
Line 255 Figure 2., line 317 Figure 3. In the Figures needs to be highlighted legends text. Now it's dificult to read.
Line 260-262 "The production of milk at the farm (including milk ingredients for FO) contributes to 82 to 88% of the climate change (CC), human health (HH) and ecosystem quality (EQ) impacts and to 33 to 43% of the fossil energy use (FEU) impacts, for all the scenarios before L&W." Whose results are here and where are they presented? Must be specified.
Line 267 "Losses and wastage (L&W) contribute to 23 to 25% of the life cycle impacts..." and Line 275-276 "Processing operations at plant contribute to less than 3% of CC, HH, EQ and between 3 to 8% of FEU impacts..." Same notes - it is not clear whose result are you providing. Must be specified.
Author Response
Dear reviewer,
Thank you very much for having taken the time to review our manuscript. Please find below our answers to your comments.
Point 1: The information you provided in lines 77-79 is not required.
Response 1: This information has been removed in the revised version.
Point 2: Line 163-164 (now line 155). "Table 1. Simulated mass balance of inputs and outputs for the CE, FO, and UF technologies before L&W". There is need explanation for the abbreviation L&W in the text. You wrote explanation in line 267.
Response 2: It has been updated in the revised version (line 155)
Point 3: Line 255 Figure 2., line 317 Figure 3. In the Figures needs to be highlighted legends text. Now it's dificult to read.
Response 3: It has been improved in the revised version.
Point 4: Line 260-262 (now lines 257-259) "The production of milk at the farm (including milk ingredients for FO) contributes to 82 to 88% of the climate change (CC), human health (HH) and ecosystem quality (EQ) impacts and to 33 to 43% of the fossil energy use (FEU) impacts, for all the scenarios before L&W." Whose results are here and where are they presented? Must be specified.
Response 4: We added the following sentence to clarify this point: « The following section details the results from Figure 2 in % » (L256)
Point 5: Line 267 (now lines 264) "Losses and wastage (L&W) contribute to 23 to 25% of the life cycle impacts..." and Line 275-276 "Processing operations at plant contribute to less than 3% of CC, HH, EQ and between 3 to 8% of FEU impacts..." Same notes - it is not clear whose result are you providing. Must be specified.
Response 5: Same than response 4
Reviewer 3 Report
I think the work is very well done. It is presented in a very appropriate way, with interesting and original figures and detailed descriptions. I would consider maybe to move part of the discussion section in the conclusions paragraph, in order to create a continuous file rouge. I would just recommend en extensive copy editing and proofreading, since I detected some very visible mistakes (such as two "paragraph 1", both introduction and materials). That is all I have to say, an acceptance after minor revision.
Author Response
Dear reviewer,
Thank you very much for having taken the time to review our manuscript. We made a few modifications based on your comments : (1) the discussion has been slightly improved; (2) The language and formatting have been revised but we will follow your recommendation and ask for an extensive copy editing and proofreading.